# POLICY OPTIMIZATION WITH $f$-DIVERGENCE REGULARIZATION

## ABSTRACT

Policy iteration is a common algorithm framework in reinforcement learning (RL) to find the optimal policy for a Markov decision process (MDP). To improve training stability and prevent catastrophic failure, researchers have developed several policy iteration algorithms based on the Kullback-Leibler (KL) divergence, such as the well-known trust region policy optimization (TRPO) and proximal policy optimization (PPO). However, these methods are limited to the KL divergence, which may not be the best choice for all environments. In this work, we generalize previous work using a more general form of divergence, the $f$-divergence, and design a new family of algorithms that can improve learning policy with theoretical improvement guarantees. Our method, $f$-divergence-regularized policy optimization ($f$RPO), can be applied to both online and offline RL settings. Empirical studies show that $f$RPO can outperform existing methods, including the commonly used KL divergence, on common benchmark problems in RL.

## 1 INTRODUCTION

In reinforcement learning (RL), an agent is deployed in an environment where it is assumed that the agent has no prior knowledge of the environment (Sutton & Barto, 2018). The agent's goal is to maximize the expected return by solving the sequential decision-making problem, where the outcomes are uncertain. This problem is often formulated as a Markov decision process (MDP) (Puterman, 2014). One approach is to calculate the expected return from the data collected by the agent, differentiate it with respect to policy parameters, and apply gradient ascent to update the agent's policy (Sutton et al., 1999). However, this approach may introduce high variance during gradient estimation, and the improved policy cannot utilize the dataset efficiently, potentially resulting in convergence to a suboptimal policy. Therefore, many previous works have focused on reducing variance when finding the optimal policy, such as using the advantage instead of the state-action value (Schulman et al., 2018) or applying approximate policy iteration algorithms to learn the optimal policy. For instance, relative entropy policy search (REPS) (Peters et al., 2010) used the advantage value and iterative constrained policy optimization to learn the optimal policy in the linear setting. Advantage weighted regression (AWR) (Peng et al., 2019) later applied a similar idea and enabled training with deep neural network models. However, they only utilized the Kullback–Leibler (KL) divergence to constrain the relative policy search step, which can be limited for some applications.

Addressing this limitation, Belousov & Peters (2019) demonstrated an alternative formulation to conduct proximal policy search using a more general form of divergence, the $f$-divergence. Instead of directly deriving the policy, they first derived the state-action distribution, then used it to learn a parametrized policy. However, these steps require more constraints to ensure that the distribution can be validly fitted into a policy, which is computationally inconvenient and not amenable to training deep models. Moreover, it lacked theoretical justifications for policy improvement.

In this paper, we derive the objective directly from the policy rather than its state-action distribution, and we arrive at an objective function that can utilize any $f$-divergence as a regularization term for the policy improvement step and is suitable for training with neural network models. In addition, we theoretically show that the policy improvement step enjoys a monotonic improvement guarantee. Finally, we demonstrate its feasibility through empirical studies, showing that it yields competitive performances compared to other methods in both the online and offline settings.

**Contributions**    The contributions of this paper are three-fold. (1) We derive a family of policy iteration algorithms, called $f$-divergence-regularized policy optimization ($f$RPO), that is suitable for training deep models in the online and offline RL settings. (2) Beyond KL divergence, we provide theoretical justifications for the policy improvement step with general $f$-divergence. (3) Our experiments show that using different $f$-divergences can have a potential benefit on the final performance, where, in most cases, some other $f$-divergences may perform better than the KL divergence.

**Paper Organization**    In the following sections, we first discuss in Sec. 2 the related work to our method and introduce some fundamental concepts. Then, we prove that using $f$-divergence regularization can have a theoretically monotonic improvement on the policy, and derive our $f$RPO algorithm in Sec. 3. Sec. 4 demonstrates empirical results obtained by different $f$-divergences of our policy optimization method, and compare them to the results from methods that use KL divergence and other recent and well-known baselines. Finally, Sec. 5 concludes the paper.

## 2    RELATED WORK

**Advantage Function**    To address the high-variance problem raised in policy gradient estimation, the early work proposed having a baseline on the objective (Williams, 1992), where it could reduce the variance of the gradient estimation. One of the methods that utilized the baseline technique and achieved great success is based on the estimation of the advantage function (Schulman et al., 2018), where the baseline is set as the value function. The advantage denotes how much better an action is compared to the value of a state. This method reduced the variance of the gradient estimation and helped the policy to converge to the optimal policy much faster. After that, many researchers such as Haarnoja et al. (2018), Mnih et al. (2016), and Li et al. (2019) have used this method and achieved great performance in practice. In our work, we also utilize this advantage as part of our objective to avoid the high-variance problem and speed up training.

**Regularized MDPs and Entropic Penalty**    In some works, researchers have added regularization terms to the original RL objectives to derive their policy objectives, as seen in works proposed by Haarnoja et al. (2018), Vieillard et al. (2021) or Yang et al. (2019). In other words, the entropy term is used in the RL objective, where to have a bonus to the original reward, by controlling the weight of the entropy term, it can incentivize certain properties such as the stochasticity of the policy. Commonly, in online setting, it may encourage the policy to explore more at the beginning, and to be more conservative as the training progresses, while in offline setting, it makes the policy more cautious about unknown actions. Compared to these works, there are some works that seek a conservative way to update the policy, where they only apply the regularization term in the policy improvement step, as done in relative entropy policy search (REPS) (Peters et al., 2010), Trust Region Policy Optimization (TRPO) (Schulman et al., 2015) or advantage weighted regression (AWR) (Peng et al., 2019). For these works, instead of encouraging or being caution for the unknown actions and states, they are trying to update the policy that not deviate too much from previous policy and the local optimal policy. By having a regularization term at the policy improvement step, it can update more stably and avoid potential crash caused by deviating too much from previous policy. In our work, we design our method in a similar way to AWR or TRPO, where we have the regularization term only in the policy improvement step.

**KL Divergence**    One of the constraint/regularization that previous works used in the policy improvement step is based on the Kullback–Leibler (KL) divergence, such as TRPO (Schulman et al., 2015). They used the KL divergence to restrict the learned policy not stepping too far from the previous policy, avoiding the performance crash due to the large updating step. Many works (Schulman et al., 2017; Abdolmaleki et al., 2018; Achiam et al., 2017; Wang et al., 2022) are established on this idea, and they also used the KL divergence as one of the constraints to update the policy. Another way to utilize the KL divergence is to derive a target policy in each policy iteration step and make the learning policy converge to this target policy, such as REPS (Peters et al., 2010) and AWR (Peng et al., 2019). Some work (Vieillard et al., 2021) has proved the feasibility of using KL divergence, or even other forms of KL divergences (Chan et al., 2022). Compared to these methods, we only use the KL divergence to approximately learn network parameters that best fit the optimal target policy in each iteration, while keeping the regularization term with a broader family of $f$-divergence for seeking the optimal target policy.

## 3 $f$-DIVERGENCE-REGULARIZED POLICY OPTIMIZATION

In this section, we introduce a family of policy iteration algorithms using $f$-divergence as regularization. Sec. 3.1 first introduces the necessary preliminaries. Then we have a theoretical analysis of the policy improvement guarantee with $f$-divergences regularization in Sec. 3.2, followed by deriving the objective that iteratively improves the learning policy, and having a target closed-form of optimal policy in Sec. 3.3. Sec. 3.4 shows how to optimize a parametrized policy given the closed-form solution. Finally, Sec. 3.5 provides some implementation details of the algorithm for practical usage.

### 3.1 PRELIMINARIES

In reinforcement learning, the core idea is to find the optimal policy in an MDP, which is defined as $(\mathcal{S}, \mathcal{A}, P, \rho_0, r, \gamma)$, where $\mathcal{S}$ denotes the state space of the environment, $\mathcal{A}$ denotes the action space, $P$ denotes the environment dynamics $P(s'|s, a)$, which is the probability of the transition from state $s \in \mathcal{S}$ to state $s' \in \mathcal{S}$ when taking an action $a \in \mathcal{A}$, $\rho_0$ denotes the initial state distribution, $r : \mathcal{S} \times \mathcal{A} \mapsto \mathbb{R}$ is the reward function, and $\gamma \in [0, 1)$ is the discount factor. An RL agent interacting with the environment is described by a policy $\pi$ that maps a state to a distribution over actions. At each time step $t$, the agent will observe state $s_t$ of the environment, take an action $a_t \sim \pi(\cdot|s_t)$ and transition to another state $s_{t+1}$, receiving a reward $r_t = r(s_t, a_t)$. The goal of the agent is to find a policy that maximizes the expected return $J(\pi)$ (Sutton & Barto, 2018):

$$\max_\pi \ J(\pi) := \mathbb{E}_{\tau \sim \pi}\left[\sum_{t=0}^\infty \gamma^t r_t\right] = \mathbb{E}_{s \sim d_\pi}\mathbb{E}_{a \sim \pi}[r(s, a)], \tag{1}$$

where the $\tau = \{(s_0, a_0, r_0), (s_1, a_1, r_1), ...\}$ is a possible trajectory that generated by the policy $\pi$, i.e., $s_0 \sim \rho_0(\cdot)$, $a_t \sim \pi(\cdot|s_t)$, $s_{t+1} \sim P(\cdot|s_t, a_t)$. $d_\pi$ represents the unnormalized discounted state frequency induced by the policy $\pi$, i.e., $d_\pi(s) = \sum_{t=0}^\infty \gamma^t \Pr(s_t = s|\pi)$, where $\Pr(s_t = s|\pi)$ denotes the probability that the agent observes the state $s$ at time step $t$ with policy $\pi$.

The state value function $V^\pi$ and state-action value function (or simply action-value function) $Q^\pi$ of a policy $\pi$ are defined as follows:

$$V^\pi(s) := \mathbb{E}_\pi\left[\sum_{i=0}^\infty \gamma^i r(s_{t+i}, a_{t+i}) \,\middle|\, s_t = s\right] \tag{2}$$

$$Q^\pi(s, a) := \mathbb{E}_\pi\left[\sum_{i=0}^\infty \gamma^i r(s_{t+i}, a_{t+i}) \,\middle|\, s_t = s, a_t = a\right] \tag{3}$$

where $a_t \sim \pi(\cdot|s_t), s_{t+1} \sim P(\cdot|s_t, a_t)$ for $t \geq 0$. As shown by Kakade & Langford (2002) and Schulman et al. (2015), we can rewrite the expected return of a policy $\pi$ in terms of the advantage value $A^{\pi'}(s, a) = Q^{\pi'}(s, a) - V^{\pi'}(s)$ of another policy $\pi'$ as

$$J(\pi) = J(\pi') + \mathbb{E}_{s \sim d_\pi}\mathbb{E}_{a \sim \pi}\left[A^{\pi'}(s, a)\right]. \tag{4}$$

However, it is hard to optimize Eq. (4) due to the complex dependency of the state distribution $d_\pi$ on the policy $\pi$. Kakade & Langford (2002) suggested a surrogate objective with respect to policy $\pi'$ as follows:

$$Z_{\pi'}(\pi) := J(\pi') + \mathbb{E}_{s \sim d_{\pi'}}\mathbb{E}_{a \sim \pi}\left[A^{\pi'}(s, a)\right], \tag{5}$$

where in this objective, the update for policy $\pi$ will not affect the state distribution $d_{\pi'}$, as it is independent of the policy $\pi$. Also, $Z_{\pi'}(\pi)$ matches $J(\pi)$ to first order (Kakade & Langford, 2002). Specifically, it matches $J(\pi')$ at $\pi = \pi'$:

$$Z_{\pi'}(\pi') = J(\pi'). \tag{6}$$

In this work, we focus on policy iteration algorithms based on Eq. (5). In iteration $k + 1$, we can treat $\pi'$ as the previous policy $\pi_k$, and $\pi$ as a learning policy. Since the term $J(\pi')$ does not involve the policy $\pi$, we eliminate it from the optimization, resulting in the following:

$$\max_\pi \ \Lambda_{\pi_k}(\pi) := \mathbb{E}_{s \sim d_{\pi_k}}\mathbb{E}_{a \sim \pi}\left[A^{\pi_k}(s, a)\right]. \tag{7}$$

Table 1: Function $f$, its convex conjugate $f_*$, and their derivatives.

| Divergence | $f(x)$ | $f'(x)$ | $(f_*)'(y)$ | $f_*(y)$ | $\mathrm{dom} f_*$ |
|---|---|---|---|---|---|
| KL | $x \log x - (x-1)$ | $\log x$ | $e^y$ | $e^y - 1$ | $\mathbb{R}$ |
| Reversed KL | $-\log x + (x-1)$ | $-\frac{1}{x} + 1$ | $\frac{1}{1-y}$ | $-\log(1-y)$ | $y < 1$ |
| Pearson $\chi^2$ | $\frac{1}{2}(x-1)^2$ | $x - 1$ | $y + 1$ | $\frac{1}{2}(y+1)^2 - \frac{1}{2}$ | $y > -1$ |
| Neyman $\chi^2$ | $\frac{(x-1)^2}{2x}$ | $-\frac{1}{2x^2} + \frac{1}{2}$ | $\frac{1}{\sqrt{1-2y}}$ | $-\sqrt{1-2y} + 1$ | $y < \frac{1}{2}$ |
| Hellinger | $2(\sqrt{x}-1)^2$ | $2 - \frac{2}{\sqrt{x}}$ | $\frac{4}{(2-y)^2}$ | $\frac{2y}{2-y}$ | $y < 2$ |

### 3.2 Monotonic Improvement Guarantee with $f$-Divergence Regularization

In this subsection, we first introduce the definition of the $f$-divergence. Next, for every policy iteration step, we establish a lower bound of the performance of a policy $\pi$ using the performance of the previous policy $\pi_k$ with the $f$-divergence. This provides theoretical justifications for learning a new policy $\pi_{k+1}$ with a monotonic improvement guarantee. Finally, we will discuss an alternative approach towards practical implementation.

Suppose that we have two distributions $P$ and $Q$, with their densities $p$ and $q$, respectively. Their $f$-divergence (Csiszár, 1963) is defined as

$$D_f(p \parallel q) := \mathbb{E}_q \left[ f\left(\frac{p}{q}\right) \right], \tag{8}$$

where the function $f : \mathbb{R}_+ \to \mathbb{R}$ should be convex with $f(1) = 0$. Some examples of $f$ are shown in Table 1. In this work, we will directly apply this to the action distributions of policies, where we treat the action distribution of a state $s$ of the learning policy $\pi$ and that of the previous policy $\pi_k$ to be the $P$ and $Q$, so the $f$-divergence respect to the $\pi(s)$ and $\pi_k(s)$ is

$$D_f(\pi(s) \parallel \pi_k(s)) = \sum_{a \in \mathcal{A}} \pi_k(a|s) f\left(\frac{\pi(a|s)}{\pi_k(a|s)}\right). \tag{9}$$

Previously, Schulman et al. (2015) showed that one can lower bound the performance of any policy $\pi$ using the total variation divergence $D_{TV}$ between two policies:

**Theorem 1** *(Schulman et al., 2015, Theorem 1) Let $C = \frac{4\gamma \max_{s,a} |A^{\pi_k}(s,a)|}{(1-\gamma)^2}$. Then*

$$J(\pi) \geq J(\pi_k) + \Lambda_{\pi_k}(\pi) - C \cdot \max_s D_{TV}\left(\pi(s) \parallel \pi_k(s)\right)^2. \tag{10}$$

Schulman et al. (2015) focused on KL divergence as a surrogate for the total variation divergence using Pinsker's inequality. However, it is possible to derive similar inequalities (Gilardoni, 2010, Theorem 3) for general $f$-divergence as $D_f(p \parallel q) \geq \frac{f''(1)}{2} D_{TV}(p \parallel q)^2$. As a result, we have the following bound

**Proposition 2** *Let $C_f = \frac{2}{f''(1)} \cdot C$. Then*

$$J(\pi) \geq J(\pi_k) + \Lambda_{\pi_k}(\pi) - C_f \cdot \max_s D_f\left(\pi(s) \parallel \pi_k(s)\right). \tag{11}$$

It generalizes the monotonic improvement guarantee of TRPO (Schulman et al., 2015) from the KL divergence to an arbitrary $f$-divergence. Specifically, for any candidate policy $\pi$, the expected return of it is lower-bounded by the sum of the previous policy's return $J(\pi_k)$ and the expected advantage $\Lambda_{\pi_k}(\pi)$, minus a penalty proportional to the $f$-divergence between $\pi$ and $\pi_k$.

Suppose that we learn a policy $\pi_{k+1}$ by maximizing the last two terms of Eq.(11), i.e.,

$$\pi_{k+1} = \arg\max_{\pi} \Lambda_{\pi_k}(\pi) - C_f \cdot \max_s D_f\left(\pi(s) \parallel \pi_k(s)\right). \tag{12}$$

Then plugging it to Eq.(11), we have

$$J(\pi_{k+1}) \geq J(\pi_k) + \Lambda_{\pi_k}(\pi_{k+1}) - C_f \cdot \max_s D_f\left(\pi_{k+1}(s) \parallel \pi_k(s)\right). \tag{13}$$

Moreover, from Eqs. (5) and (7), we know that this is equivalent to

$$J(\pi_{k+1}) \geq Z_{\pi_k}(\pi_{k+1}) - C_f \cdot \max_s D_f\left(\pi_{k+1}(s) \parallel \pi_k(s)\right). \tag{14}$$

From Eq. (6) and the fact that $D_f(\pi_k(s)\|\pi_k(s)) = 0, \forall s$, we have

$$J(\pi_k) = Z_{\pi_k}(\pi_k) = Z_{\pi_k}(\pi_k) - C_f \cdot \max_s D_f\left(\pi_k(s) \parallel \pi_k(s)\right) \tag{15}$$

Subtracting Eq. (15) from Eq. (14) gives

$$J(\pi_{k+1}) - J(\pi_k) \geq \left[Z_{\pi_k}(\pi_{k+1}) - C_f \cdot \max_s D_f\left(\pi_{k+1}(s) \parallel \pi_k(s)\right)\right]$$

$$- \left[Z_{\pi_k}(\pi_k) - C_f \cdot \max_s D_f\left(\pi_k(s) \parallel \pi_k(s)\right)\right] \tag{16}$$

$$= \left[\Lambda_{\pi_k}(\pi_{k+1}) - C_f \cdot \max_s D_f\left(\pi_{k+1}(s) \parallel \pi_k(s)\right)\right]$$

$$- \left[\Lambda_{\pi_k}(\pi_k) - C_f \cdot \max_s D_f\left(\pi_k(s) \parallel \pi_k(s)\right)\right] \tag{17}$$

where the last equality uses Eqs. (5) and (7). Finally, according to Eq. (12), $\pi_{k+1}$ is the maximizer, so the last equation is greater than or equal to zero, i.e.,

$$J(\pi_{k+1}) - J(\pi_k) \geq 0, \tag{18}$$

which is a monotonic improvement guarantee for policy iteration.

Although Eq. (12) provides monotonic improvement, computing the maximum $f$-divergence over all states is challenging in general. Therefore, we resort to an average divergence instead. Specifically, we use $\mathbb{E}_{s \sim d_{\pi_k}}[D_f(\pi(s) \parallel \pi_k(s))]$ to approximate $\max_s D_f(\pi(s) \parallel \pi_k(s))$, with an appropriately chosen temperature $\tau$, similar to Schulman et al. (2015). In the next subsection, we will show how to solve this altered objective.

### 3.3 TARGET POLICY CLOSED-FORM SOLUTION

Since the surrogate objective Eq. (7) is only accurate locally, we need to optimize it without deviating too much from the previous policy, which leads to a policy iteration algorithm. Motivated by Sec. 3.2, we add an average $f$-divergence regularization term to measure the difference between the new and previous policies in each policy iteration. A temperature parameter $\tau$ controls the strength of this regularization. Thus, we have the following objective[1]:

$$\max_\pi \Lambda_{\pi_k}(\pi) - \tau\mathbb{E}_{s \sim d_{\pi_k}}[D_f(\pi(s) \parallel \pi_k(s))] \tag{19}$$

$$\text{s.t.} \sum_a \pi(a|s) = 1 \quad \forall s \in \mathcal{S} \tag{20}$$

$$\pi(a|s) \geq 0 \quad \forall (s,a) \in \mathcal{S} \times \mathcal{A} \tag{21}$$

As a special case, advantage weighted regression (AWR) (Peng et al., 2019) applied the KL divergence as the regularization, whereas here we use a more general family of divergences. The constraint (20) ensures that the sum of all action probabilities of every state is one. The constraint (21) enforces non-negativity for the policy probability for each state-action pair. Then, we transform this objective into Lagrangian form by substituting the Eq. (7) to Eq. (19). Since the state sampling distribution $d_{\pi_k}$ does not depend on the policy $\pi$, we can optimize this objective independently for each state. For a given state $s$, the optimal solution is given by (details in Appendix A.1)

$$\pi^*(a|s) = \pi_k(a|s)f'_*\left(\frac{A^{\pi_k}(s,a) - \lambda(s) + \kappa(s,a)}{\tau}\right), \tag{22}$$

where $\lambda(s) \in \mathbb{R}$ and $\kappa(s,a) \geq 0$ are the dual variables and $f_*$ is the convex conjugate of $f$.

---

[1] Having the regularization term $\tau\mathbb{E}_{s \sim d_{\pi_k}}[D_f(\pi(s) \parallel \pi_k(s))]$ is equivalent to having a constraint on the $f$-divergence of the form $\mathbb{E}_{s \sim d_{\pi_k}}[D_f(\pi(s) \parallel \pi_k(s))] \leq \epsilon$ for some $\epsilon$.

### 3.4 OPTIMIZING A PARAMETRIZED POLICY

Despite knowing the optimal target policy $\pi^*$ in closed-form as in Eq.(22), it may not be directly useable/implementable. Therefore, we learn a parametrized policy $\pi_\theta$ to approximate it. To do so, we minimize the KL divergence between the optimal policy and the parametrized one as follows:

$$\min_\theta \quad \mathbb{E}_{s \sim d_{\pi_k}}[D_{KL}(\pi^*(s) \parallel \pi_\theta(s))] \tag{23}$$

$$\iff \min_\theta \quad \mathbb{E}_{s \sim d_{\pi_k}} \sum_a \pi^*(a|s) \log \pi^*(a|s) - \pi^*(a|s) \log \pi_\theta(a|s) \tag{24}$$

$$\iff \max_\theta \quad \mathbb{E}_{s \sim d_{\pi_k}} \sum_a \pi^*(a|s) \log \pi_\theta(a|s) \tag{25}$$

$$\iff \max_\theta \quad \mathbb{E}_{s \sim d_{\pi_k}} \sum_a \pi_k(a|s) f'_* \left( \frac{A^{\pi_k}(s,a) - \lambda(s) + \kappa(s,a)}{\tau} \right) \log \pi_\theta(a|s) \tag{26}$$

$$\iff \max_\theta \quad \mathbb{E}_{s \sim d_{\pi_k}} \mathbb{E}_{a \sim \pi_k} \left[ \log \pi_\theta(a|s) f'_* \left( \frac{A^{\pi_k}(s,a) - \lambda(s) + \kappa(s,a)}{\tau} \right) \right] \tag{27}$$

This provides an objective for finding an approximate solution to $\pi^*$ since the advantage of the current policy $\pi_k$ can be estimated from data. However, it is not yet easily implementable since the dual variables $\lambda, \kappa$ remain to be identified. In the next subsection, we will develop a practical algorithm for implementation.

### 3.5 A PRACTICAL IMPLEMENTATION

In this subsection, we show that even though one could find the dual variables by solving the dual optimization problem, it is not very stable or efficient. As a result, we propose an modification to the implementation that works well in practice.

The dual problem to Eq.(19), given a state $s$, is (details in Appendix A.2)

$$\min_{\lambda, \kappa} \quad g(\lambda, \kappa) = \tau \mathbb{E}_{\pi_k} \left[ f_* \left( \frac{A^{\pi_k}(s,a) - \lambda(s) + \kappa(s,a)}{\tau} \right) \right] + \lambda(s) \tag{28}$$

$$\text{s.t.} \quad \kappa(s,a) \geq 0, \quad \forall (s,a) \in \mathcal{S} \times \mathcal{A} \tag{29}$$

$$\frac{A^{\pi_k}(s,a) - \lambda(s) + \kappa(s,a)}{\tau} \in \text{dom}(f_*), \quad \forall (s,a) \in \mathcal{S} \times \mathcal{A} \tag{30}$$

One could solve this problem in each policy iteration and find the optimal dual variables for the primal training. We attempted this approach and monitored the value of each dual variable. We first observed that $\kappa$ is almost always zero for a diverse range of environments and $f$ functions that we tried, so for simplicity, we ignore it and focus on learning $\lambda$ in the subsequent discussion.

Based on the constraint (30), we can see that one purpose of $\lambda(s)$ is, for a given state $s$, to shift the original advantage values of $s$ to make sure that they are within the domain of $f_*$ (see Table 1). However, for continuous state and action spaces, finding such a $\lambda(\cdot)$ function is challenging as there might be some unobserved states or actions that violate the constraint. Therefore, we further simplify $\lambda(s)$ to a scalar $\lambda$ that is applied to all examples in a mini-batch during training, allowing a small portion to violate the constraint due to numerical reasons. If an example in a mini-batch violates the constraint, the $\lambda$ should be adjusted so that the LHS of Eq.(30) gets closer to the boundary value (as if it is being "clipped" back to the domain). We monitor the clipping ratio $\mathcal{C}$ (number of examples being clipped over number of all examples in a mini-batch) and aim to adjust $\lambda$ so that it approaches the desired target ratio $\mathcal{C}_{target} \in [0,1]$. Inspired by Ziegler et al. (2020), we adjust $\lambda$ as follows:

$$\lambda = \lambda + \beta \left( \frac{\mathcal{C} - \mathcal{C}_{target}}{\mathcal{C}_{target}} \right), \tag{31}$$

where $\beta > 0$ is the step size. The goal of this equation is to stabilize the value of $\lambda$ by making the clipping ratio $\mathcal{C}$ converge to the target clipping ratio $\mathcal{C}_{target}$. When $\mathcal{C}$ is larger than $\mathcal{C}_{target}$, the term $\frac{\mathcal{C} - \mathcal{C}_{target}}{\mathcal{C}_{target}}$ will be positive, then $\lambda$ will increase, and the term $y := \frac{A(s,a) - \lambda + \kappa(s,a)}{\tau}$ in Eq.(27) and Eq.(28) will be smaller for each state-action pair. Based on Table 1, when $y$ becomes smaller,

almost all the divergences will have more state-action pairs to stay in the domain of $f_*$ except for Pearson. For Pearson, we flip the sign of Eq.(31) from plus to minus. In other words, Eq.(31) is applied when the domain has an upper bound and the sign flipped when it has a lower-bounded domain. With this, all divergences will have lower cutting ratio $\mathcal{C}$ as $\lambda$ increase, which means we have $\mathcal{C}$ converge to $\mathcal{C}_{target}$. It works similarly when $\mathcal{C}$ is lower than $\mathcal{C}_{target}$, where $\lambda$ will decrease, and more state-action pairs will be clipped. $\mathcal{C}$ will then increase and approach $\mathcal{C}_{target}$. The reason we have a normalized difference (i.e., divided by $\mathcal{C}_{target}$) is to speed up convergence when $\mathcal{C}$ is too far from $\mathcal{C}_{target}$. Also, we don't want the $\mathcal{C}$ bouncing back and force around the $\mathcal{C}_{target}$, so we set the $\beta$ to $0.1$ in our work to stabilize training when $\mathcal{C}$ is close to $\mathcal{C}_{target}$.

Algorithm 1 summarizes our method, $f$-divergence-regularized policy optimization ($f$RPO). Although our method is designed for online RL, it can be embedded into other offline algorithms for policy training. For example, implicit Q-learning (IQL) (Kostrikov et al., 2022) is an offline RL method that approximates the max operator using expectile regression when training the state value function from a given dataset. In our offline experiments, we will show that one can replace IQL's policy extraction step with our $f$RPO to conduct offline RL.

More specifically, in addition to the policy $\pi_\theta$, we also need to estimate the state-value function $V$ and action-value function $Q$ in order to calculate the advantage. Suppose that we have a state-value network $V_\phi$ parametrized by $\phi$ and an action-value network $Q_\psi$ parametrized by $\psi$. As suggested by Twin Delayed DDPG (TD3) (Fujimoto et al., 2018), we use clipped double-Q learning in order to avoid the overestimation of the $Q$ value. Thus, we have two copies of action value network $Q_{\psi_1}$ and $Q_{\psi_2}$. We use hats to denote target network parameters (e.g., $\hat{\psi}_1$), which are updated slowly using exponential moving average. We estimate the advantage value as follows

$$A^{\pi_\theta}(s,a) \approx \min_{i=1,2} Q_{\hat{\psi}_i}(s,a) - V_\phi(s). \tag{32}$$

In the online setting, we regress the state-value network towards the Monte-Carlo return $G(s) = \sum_{t=0} \gamma^t r_t$ with $s_0 = s$, and for the action-value networks, we use the bootstrap target

$$\mathcal{L}_V(\phi) = \mathbb{E}_{s \sim \mathcal{D}} \left[ \left( V_\phi(s) - G(s) \right)^2 \right] \tag{33}$$

$$\mathcal{L}_Q(\psi_1, \psi_2) = \mathbb{E}_{(s,a,s') \sim \mathcal{D}} \left[ \left( Q_{\psi_1}(s,a) - \left( r(s,a) + \gamma V_{\hat{\phi}}(s') \right) \right)^2 \right.$$
$$\left. + \left( Q_{\psi_2}(s,a) - \left( r(s,a) + \gamma V_{\hat{\phi}}(s') \right) \right)^2 \right] \tag{34}$$

where $\mathcal{D}$ is the replay buffer[2]. In the offline setting, we keep the same loss function (34) for $Q$ while following the loss function of $V$ by IQL (Kostrikov et al., 2022):

$$\mathcal{L}_V(\phi) = \mathbb{E}_{(s,a) \sim \mathcal{D}} \left[ L_2^\rho \left( \min_{i=1,2} Q_{\hat{\psi}_i}(s,a) - V_\phi(s) \right) \right] \tag{35}$$

where $L_2^\rho(u) = |\rho - \mathbb{1}(u < 0)| u^2$ is an asymmetric loss for expectile regression and $\mathcal{D}$ is the offline dataset instead.

## 4 EXPERIMENTS

This section demonstrates the applicability of our $f$RPO in online and offline settings. We aim to show that one can learn the policy using a diverse set of regularization based on different $f$-divergences, rather than relying solely on the KL divergence, as done in AWR (Peng et al., 2019). We will first demonstrate the performance in online environments in Sec.4.1. Then, we will demonstrate the performance of using different $f$-divergence functions as the policy extraction method in an offline setting in Sec.4.2. Additional results and details can be found in Appendix B.

### 4.1 $f$RPO IN ONLINE SETTING

For online environments, we use several OpenAI Gym (Brockman et al., 2016) environments that are commonly used in the RL literature We compare $f$RPO with other baselines. One of the main

---

[2]In theory, the samples should be on-policy. However, as shown in AWR (Peng et al., 2019), one may still use samples from the replay buffer for training.

---

**Algorithm 1** $f$-Divergence-Regularized Policy Optimization ($f$RPO)

---

**Require:** Policy network parameters $\theta$, state value network parameters $\phi$, action value network parameters $\psi_1, \psi_2$, target network parameters $\hat{\phi}, \hat{\psi}_1, \hat{\psi}_2$, target clipping ratio $\mathcal{C}_{target}$, Lagrangian dual variable $\lambda$, temperature $\tau$

**Require:** An empty replay buffer $\mathcal{D}$ if online or an offline dataset $\mathcal{D}$ if offline, number of samples $n$ collected by the policy after each update if online

1: Initialize networks' parameters $\theta, \phi, \hat{\phi}, \psi_1, \psi_2, \hat{\psi}_1, \hat{\psi}_2$, Lagrangian dual variable $\lambda$
2: **repeat**
3:    **if** online **then**
4:        Collect $n$ samples from the environment using $\pi_\theta$ and put them into the replay buffer $\mathcal{D}$
5:    **end if**
6:    **for** each gradient step **do**
7:        Sample mini-batches $B = \{(s, a, s', r)\}$ from $\mathcal{D}$
8:        Estimate the advantage value using Eq.(32)
9:        Calculate $y = \frac{A^{\pi_\theta}(s,a) - \lambda}{\tau}$
10:        Clip $y$ based on domain in Table 1 and record clipping ratio $\mathcal{C}$
11:        Update $\lambda$ value using Eq.(31)
12:        Update $\phi$ by gradient descent using Eq.(33) if online, or Eq.(35) if offline
13:        Update $\psi_1, \psi_2$ using Eq.(34)
14:        Update target networks' parameters with exponential moving average
15:        Update $\theta$ by gradient ascent using Eq.(27)
16:    **end for**
17: **until** convergence

---

baselines is AWR (Peng et al., 2019), as it only focuses on the KL divergence to obtain the surrogate objective, whereas ours are derived in terms of any $f$-divergence. Additionally, we compare to TRPO (Schulman et al., 2015) and PPO (Schulman et al., 2017), which use KL divergence to be the surrogate objective's regularization and they are on-policy. We also selected some off-policy baselines such as TD3 (Fujimoto et al., 2018) and SAC (Haarnoja et al., 2018).

We set the target clipping ratio $\mathcal{C}_{target}$ for $f$-divergences except for the KL divergence to 0.15. For all $f$-divergences, we set the temperature $\tau = 1$, the default value in AWR's original implementation. Ablation studies of the hyper-parameters is discussed in Appendix B.1.1. We evaluated the methods by setting the environment's seed to 1, 2, and 3, and averaged the last 5 evaluations of every seed when the training curve was stable. The results are shown in Table 2 and Fig. 1.

From Table 2, we can observe that Pearson and Neyman divergences can yield competitive results compared to the KL divergence. On the one hand, the Pearson divergence can outperform other $f$-divergences in most of the environments, including the KL divergence used in AWR. indicating that the Pearson divergence could be more suitable for these online environments. On the other hand, the reversed KL and Hellinger do not perform very well, where their training curves show that they have reached stable performance and are still not as good as other divergences.

We used the results of other baselines from Peng et al. (2019) and show the comparison in Table 2. It demonstrates that compared to other methods, our method using Pearson can beat both TRPO and PPO in all the environments. Compared to TD3 and SAC, our method have competitive results among all the environments, especially in the Hopper-v2 and Walker2d-v2 where our method with Pearson has the highest performances.

### 4.2 $f$RPO IN OFFLINE SETTING

To further prove that our method can work well in the offline setting, we use the D4RL (Fu et al., 2021) datasets to train our agent. It consists of offline samples from Hopper, Walker-2D, and Halfcheetah with different versions, indicating qualities of the samples, such as "medium" or "expert". We use them to show that our method can work well in variety of settings.

We compare $f$RPO using the reversed KL divergence to other related offline baselines, such as implicit Q-learning (IQL) (Kostrikov et al., 2022) and AWAC (Nair et al., 2021). These methods use KL divergence as the regularization term in objective, similar to AWR. We also use some other baselines that have the state-of-the-art performance in the offline setting, such as conservative Q-

Table 2: Different $f$-divergences in online setting compare to other baselines. The highest performance is bolded, and the second highest performance is underlined. The performances with a p-value $\leq 0.05$ according to one-sided $t$-test—where the hypothesis states that the mean performance of AWR is lower than that of the compared $f$-divergence method—are highlighted in red. All baselines' results are from AWR paper Peng et al. (2019).

| Environment | TRPO | PPO | TD3 | SAC | AWR (KL) | rKL | Neyman | Hellinger | Pearson |
|---|---|---|---|---|---|---|---|---|---|
| Ant-v2 | 2901 | 1161 | 4285 | **5909** | $5295 \pm 153$ | $1199 \pm 375$ | $4509 \pm 626$ | $1125 \pm 138$ | $\underline{5894} \pm 583$ |
| Halfcheetah-v2 | 3302 | 4309 | 4305 | **9297** | $9290 \pm 178$ | $6912 \pm 695$ | $8904 \pm 314$ | $5110 \pm 882$ | $8780 \pm 517$ |
| Hopper-v2 | 1880 | 1391 | 935 | $\underline{2769}$ | $1915 \pm 744$ | $2386 \pm 607$ | $2525 \pm 914$ | $2069 \pm 1083$ | $\mathbf{2862} \pm 854$ |
| Humanoid-v2 | 552 | 695 | 81 | **8048** | $5507 \pm 207$ | $651 \pm 170$ | $5824 \pm 203$ | $556 \pm 24$ | $6302 \pm 113$ |
| Walker2d-v2 | 2765 | 2617 | 4212 | 5805 | $6285 \pm 703$ | $5226 \pm 800$ | $5269 \pm 1004$ | $4163 \pm 1205$ | $\mathbf{6681} \pm 569$ |
| Total | 11400 | 10173 | 13818 | **31828** | 28292 | 16374 | 27031 | 13023 | $\underline{30519}$ |

Table 3: Reversed KL in offline datasets compare to other baselines. The highest performance is bolded, and the second highest performance is underlined. The performances with a p-value $\leq 0.05$ according to one-sided $t$-test—where the hypothesis states that the mean performance of IQL is lower than that of the compared $f$-divergence method—are highlighted in red. CQL/AWAC/IQL results are from Kostrikov et al. (2022), DT results are from Chen et al. (2021), GDT/StARformer results are from Hu et al. (2023). Ho stands for Hopper, Wa stands for Walker2d, Ha stands for Halfcheetah, m stands for medium, e stands for expert, and mr stands for medium-replay.

| Dataset | CQL | AWAC | DT | GDT | StARformer | IQL (KL) | Pearson | Neyman | Hellinger | rKL |
|---|---|---|---|---|---|---|---|---|---|---|
| Ha-m | 44.5 | 43.5 | 42.6 | 42.9 | 42.9 | $47.4 \pm 0.5$ | $46.3 \pm 0.3$ | $47.6 \pm 0.6$ | $\mathbf{48.5} \pm 0.4$ | $48.3 \pm 0.7$ |
| Ho-m | 58.5 | 57.0 | **67.6** | $\underline{65.8}$ | 59.5 | $55.8 \pm 4.5$ | $54.5 \pm 5.0$ | $54.9 \pm 5.2$ | $55.3 \pm 3.8$ | $57.5 \pm 4.5$ |
| Wa-m | 72.5 | 72.4 | 74.0 | $\underline{77.8}$ | 73.8 | $\mathbf{78.4} \pm 4.3$ | $77.4 \pm 5.7$ | $75.5 \pm 5.7$ | $75.9 \pm 6.5$ | $72.8 \pm 6.7$ |
| Ha-m-e | 91.6 | 42.8 | 86.8 | 92.4 | **93.7** | $88.5 \pm 6.4$ | $\underline{93.4} \pm 2.6$ | $91.3 \pm 4.4$ | $86.5 \pm 7.3$ | $84.0 \pm 6.7$ |
| Ho-m-e | 105.4 | 55.8 | 107.6 | $\underline{110.9}$ | **111.1** | $64.4 \pm 34.2$ | $107.0 \pm 5.8$ | $86.8 \pm 15.4$ | $67.6 \pm 18.8$ | $102.1 \pm 8.7$ |
| Wa-m-e | 108.8 | 74.5 | 108.1 | $\underline{109.3}$ | 109.0 | $\mathbf{109.9} \pm 0.9$ | $108.9 \pm 2.0$ | $109.1 \pm 1.8$ | $108.3 \pm 2.9$ | $109.0 \pm 2.2$ |
| Ha-mr | **45.5** | 40.5 | 36.6 | 39.9 | 36.8 | $\underline{44.0} \pm 0.8$ | $42.3 \pm 1.5$ | $43.1 \pm 1.5$ | $44.0 \pm 1.2$ | $43.8 \pm 1.2$ |
| Ho-mr | **95.0** | 37.2 | 82.7 | 81.6 | 29.2 | $64.4 \pm 7.2$ | $55.3 \pm 6.4$ | $66.9 \pm 6.5$ | $76.2 \pm 9.2$ | $\underline{87.9} \pm 7.4$ |
| Wa-mr | **77.2** | 27.0 | 66.6 | $\underline{74.8}$ | 39.8 | $72.5 \pm 10.3$ | $62.1 \pm 8.1$ | $69.3 \pm 9.4$ | $68.2 \pm 7.8$ | $72.1 \pm 8.5$ |
| Total | **699.0** | 450.7 | 672.6 | $\underline{695.4}$ | 595.8 | 625.3 | 647.2 | 644.5 | 630.5 | 677.5 |

learning (CQL) (Kumar et al., 2020), Decision Transformer (DT) (Chen et al., 2021), GDT (Hu et al., 2023) and StARformer (Shang et al., 2022).

We set the regularization temperature $\tau$ for the $f$-divergences except for the KL divergence to $\frac{1}{15}$, and set the target clipping ratio $\mathcal{C}_{target}$ to 0.15. The detailed discussion of setting hyper-parameters is in Appendix B.1.2. For KL divergence, which is the original implementation of IQL, we set $\tau$ to $\frac{1}{3}$, as suggested in its paper (Kostrikov et al., 2022). For all other $f$-divergences, we set the expectile level of the IQL to 0.7, the same value used in the IQL paper. We averaged the last 10 evaluations of every seed to obtain the final performance in Table 3.

Table 3 shows that KL divergence does not have the highest performance most of the time among the $f$-divergence family, which means it may not be the best choice of $f$-divergence in many offline datasets. At the same time, other $f$-divergences can yield competitive results. Besides, all other $f$-divergences have better performance in overall/total performance than KL divergence. Additionally, we can observe that unlike the online setting, where reversed KL and Helliger did not perform very well, for most of the offline datasets, all $f$-divergences have reasonable performances.

We also compared with other baselines as shown in the left portion of Table 3. It demonstrates that compared to other baselines, our method outperforms all baselines on the HalfCheetah-medium dataset. Meanwhile, compared to transformer-based methods' performance in medium-expert datasets and the CQL performance in medium-replay datasets, our method also have competitive results where the performance is close to the highest ones.

## 4.3 MODE-COVERING AND MODE-SEEKING

The different performance of various $f$-divergence may be explained from the perspective of mode-seeking versus mode-covering. As noted by Li & Farnia (2023), different $f$-divergences exhibit different tendencies towards mode-seeking (concentrating on a high value region or mode) or mode-

covering (attempting to cover multiple modes). Mode-covering encourages greater exploration, whereas mode-seeking promotes more conservative behaviour by exploiting regions that appear promising. As seen from Figure 1 of Li & Farnia (2023), Pearson divergence shows the strongest mode-covering tendency, followed by KL divergence. In contrast, reverse KL and Neyman divergences exhibit increasing degrees of mode-seeking behaviour, with Neyman divergence being the most mode-seeking overall. Hellinger divergence lies in the middle, exhibiting a balanced degree of mode-covering and mode-seeking.

In our online and offline experiments, these theoretical characteristics are consistent with empirical performance. Divergences with stronger mode-covering tendencies (Pearson, KL) achieve superior results in online environments, where exploration is crucial. In contrast, divergences with stronger mode-seeking tendencies (reverse KL, Neyman) perform well on offline datasets, where conservative behaviour mitigates extrapolation error. These observations suggest that the choice of $f$-divergence should align with the nature of the task: exploration-oriented divergences are preferable in online settings such as OpenAI Gym (Brockman et al., 2016), while conservative divergences are more suitable for offline datasets such as D4RL (Fu et al., 2021). Our method allows the $f$-divergence to be switched easily, enabling practitioners to adjust the balance between mode-covering and mode-seeking to best fit the task at hand.

## 5 CONCLUSION

In conclusion, in this work, we proposed $f$-divergence-regularized policy optimization ($f$RPO), a policy iteration algorithm that searches for a new policy within the proximity of the previous policy where the proximity is measured based on $f$-divergence. It was derived from a surrogate objective with a closed-form solution, and we developed a practical algorithm to learn the policy using flexible function approximators such as deep neural network models. The algorithm is theoretically motivated with a monotonic improvement guarantee, and it is suitable for RL in the online and offline settings. Finally, we demonstrate the feasibility of applying different $f$-divergences for online environments and offline datasets, show the potential improvements over existing methods, and discuss how various $f$-divergences can be appropriately matched to different types of environments or datasets.

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

## A DERIVATION

### A.1 OPTIMAL SOLUTION TO THE SURROGATE

To obtain the Lagrangian of (19), we first introduce our Lagrangian parameters, $\lambda(s)$ and $\kappa(s,a)$, where the $\lambda(s)$ corresponding to the constraint (20), and the $\kappa(s,a)$ corresponding to the non-negativity constraint (21). Because it is state-independent, we could first have

$$\mathcal{L}(\pi, \lambda, \kappa) = \Lambda_{\pi_k}(\pi) - \tau \left[ D_f(\pi(s) \parallel \pi_k(s)) \right]$$
$$+ \lambda(s) \left( 1 - \sum_a \pi(a|s) \right) + \sum_a \kappa(s,a)\pi(a|s). \tag{36}$$

Then we substitute our surrogate objective (7) to this, for a given state $s$, we get

$$\mathcal{L}(\pi, \lambda, \kappa) = \mathbb{E}_{a \sim \pi}[A^{\pi_k}(s, a)] - \tau[D_f(\pi(s) \parallel \pi_k(s))]$$

$$+ \lambda(s) \left(1 - \sum_a \pi(a|s)\right) + \sum_a \kappa(s, a)\pi(a|s) \tag{37}$$

$$= \sum_a \pi(a|s)A^{\pi_k}(s, a) - \tau[D_f(\pi(s) \parallel \pi_k(s))]$$

$$- \lambda(s) \left(\sum_a \pi(a|s) - 1\right) + \sum_a \kappa(s, a)\pi(a|s) \tag{38}$$

$$= \sum_a \pi(a|s)A^{\pi_k}(s, a) - \tau \sum_a \pi_k(a|s)f\left(\frac{\pi(a|s)}{\pi_k(a|s)}\right)$$

$$- \lambda(s) \left(\sum_a \pi(a|s) - 1\right) + \sum_a \kappa(s, a)\pi(a|s) \tag{39}$$

$$= \sum_a \pi(a|s)A^{\pi_k}(s, a) - \tau \sum_a \pi_k(a|s)f\left(\frac{\pi(a|s)}{\pi_k(a|s)}\right)$$

$$- \lambda(s) \left(\sum_a \pi(a|s) - 1\right) + \sum_a \kappa(s, a)\pi(a|s). \tag{40}$$

Then we calculate the derivative of this w.r.t. the policy and obtain

$$\frac{\partial \mathcal{L}}{\partial \pi(a|s)} = A^{\pi_k}(s, a) - \tau f'\left(\frac{\pi(a|s)}{\pi_k(a|s)}\right) \cdot \frac{\pi_k(a|s)}{\pi_k(a|s)} - \lambda(s) + \kappa(s, a). \tag{41}$$

By setting it to zero, we have the optimal solution

$$A^{\pi_k}(s, a) - \tau f'\left(\frac{\pi(a|s)}{\pi_k(a|s)}\right) - \lambda(s) + \kappa(s, a) = 0 \tag{42}$$

$$\iff \tau f'\left(\frac{\pi(a|s)}{\pi_k(a|s)}\right) = A^{\pi_k}(s, a) - \lambda(s) + \kappa(s, a) \tag{43}$$

$$\iff f'\left(\frac{\pi(a|s)}{\pi_k(a|s)}\right) = \frac{A^{\pi_k}(s, a) - \lambda(s) + \kappa(s, a)}{\tau} \tag{44}$$

$$\iff \frac{\pi(a|s)}{\pi_k(a|s)} = f'_*\left(\frac{A^{\pi_k}(s, a) - \lambda(s) + \kappa(s, a)}{\tau}\right) \tag{45}$$

$$\iff \pi(a|s) = \pi_k(a|s)f'_*\left(\frac{A^{\pi_k}(s, a) - \lambda(s) + \kappa(s, a)}{\tau}\right). \tag{46}$$

Here we use the fact that $(f')^{-1} = f'_*$ where $f_*$ is the convex conjugate of $f$.

## A.2 DUAL VARIABLE OPTIMIZATION

First we let $f'_*(s, a) = f'_*\left(\frac{A^{\pi_k}(s,a) - \lambda(s) + \kappa(s,a)}{\tau}\right)$ in the following derivation for simplicity. Then, based on Eq.(22), we have the optimal policy $\pi^*$ expression as

$$\pi^*(a|s) = \pi_k(a|s)f'_*(s, a). \tag{47}$$

Table 4: The summation of all environments' average performance for $f$-divergences, with $\tau = 1.0$

| $\mathcal{C}_{target}$ | 0.1 | 0.15 | 0.2 | 0.25 |
|---|---|---|---|---|
| Reversed KL | 11246 | 16233 | 17873 | 16625 |
| Pearson | 29258 | 30516 | 31299 | 31198 |
| Neyman | 28428 | 27028 | 26260 | 27358 |
| Hellinger | 10720 | 13022 | 11629 | 11071 |

After that, we plug in Eq.(47) to the Lagrangian Eq.(40), and get the objective of dual variables:

$$g(\lambda, \kappa) = \sum_a \pi_k(a|s) f'_*(s,a) A^{\pi_k}(s,a) - \tau \sum_a \pi_k(a|s) f\left(f'_*(s,a)\right)$$

$$- \lambda(s)\left(\sum_a \pi_k(a|s) f'_*(s,a) - 1\right) + \sum_a \kappa(s,a)\pi_k(a|s)f'_*(s,a) \tag{48}$$

$$= \mathbb{E}_{\pi_k}\left[f'_*(s,a)A^{\pi_k}(s,a) - \tau f\left(f'_*(s,a)\right) - \lambda(s)f'_*(s,a) + \lambda(s) + \kappa(s,a)f'_*(s,a)\right] \tag{49}$$

$$= \tau\mathbb{E}_{\pi_k}\left[\frac{A^{\pi_k}(s,a) - \lambda(s) + \kappa(s,a)}{\tau} \cdot f'_*(s,a) - f\left(f'_*(s,a)\right)\right] + \lambda(s). \tag{50}$$

For a given action $a$, it can be seen that $x := f'_*(s,a)$ and $x_* := \frac{A^{\pi_k}(s,a)-\lambda(s)+\kappa(s,a)}{\tau}$ are dual with each other (i.e., $x = f'_*(x_*)$ and $x_* = f'(x)$ since $(f')^{-1} = f'_*$). Due to the definition of the convex conjugate, we have $f_*(x_*) = x \cdot x_* - f(x)$ when $x$ and $x_*$ are dual with each other. Therefore, the dual objective can be simplified to

$$g(\lambda, \kappa) = \tau\mathbb{E}_{\pi_k}\left[f_*\left(\frac{A^{\pi_k}(s,a) - \lambda(s) + \kappa(s,a)}{\tau}\right)\right] + \lambda(s) \tag{51}$$

Finally, we consider the range of $x$, because $x$ also equals to $\pi^*(a|s)/\pi_k(a|s) \geq 0$, which means $x_*$ has to be in the set $\{f'(x)|x \geq 0\}$, or the domain of $f_*$. As a result, the dual problem is given by

$$\min_{\lambda,\kappa} \quad g(\lambda, \kappa) = \tau\mathbb{E}_{\pi_k}\left[f_*\left(\frac{A^{\pi_k}(s,a) - \lambda(s) + \kappa(s,a)}{\tau}\right)\right] + \lambda(s) \tag{52}$$

$$\text{s.t.} \quad \kappa(s,a) \geq 0, \quad \forall(s,a) \in \mathcal{S} \times \mathcal{A} \tag{53}$$

$$\frac{A^{\pi_k}(s,a) - \lambda(s) + \kappa(s,a)}{\tau} \in \text{dom}(f_*), \quad \forall(s,a) \in \mathcal{S} \times \mathcal{A}. \tag{54}$$

# B  EXPERIMENT DETAILS

## B.1  ABLATION STUDIES

### B.1.1  EVALUATION OF ONLINE HYPER-PARAMETERS

In online environments, we set the $\tau$ value to the AWR's default setting of 1, and we only need to find a suitable universal $\mathcal{C}_{target}$. We record and average the last 10 evaluations among seeds 1, 2, and 3 for each online environment and sum up all the average results from each environment to determine the $\mathcal{C}_{target}$. We test the $\mathcal{C}_{target}$ with values 0.1, 0.15, 0.2, and 0.25; the results are in Table 4. It demonstrates that when $\mathcal{C}_{target}$ is 0.1, both reversed KL and Hellinger divergences perform worse than other settings, and when $\mathcal{C}_{target}$ is 0.15, 0.2, and 0.25, reversed KL, Pearson and Neyman divergences' performances are not changing too much. However, the Hillinger divergence's performance is worse when $\mathcal{C}_{target}$ is 0.2 and 0.25. Thus, we decided to set $\mathcal{C}_{target} = 0.15$, where we could have a better trade-off between different $f$-divergences' performances.

### B.1.2  EVALUATION OF OFFLINE HYPER-PARAMETERS

In offline setting, besides the clipping ratio $\mathcal{C}_{target}$, we also need to find a proper temperature $\tau$. We decide to choose Hellinger and Neyman divergences to find the universal hyper-parameters. For the evaluation, we give these two divergences different combination of $\tau$ and $\mathcal{C}_{target}$, and we record and

Table 5: The summation of all datasets' average performance for Neyman and Hellinger

| $\mathcal{C}_{target}$ | 0.1 | | 0.15 | | 0.2 | |
|---|---|---|---|---|---|---|
| Inverse $\tau$ | 15 | 20 | 15 | 20 | 15 | 20 |
| Neyman | 1277.7 | 1284.5 | 1264.0 | 1261.5 | 1267.7 | 1231.4 |
| Hellinger | 1148.1 | 1174.0 | 1243.7 | 1251.1 | 1263.3 | 1222.1 |

Table 6: The summation of all datasets' average performance for Neyman and Hellinger, with setting $\mathcal{C}_{target}$ to 0.15

| Inverse $\tau$ | 10 | 15 | 20 | 25 |
|---|---|---|---|---|
| Neyman | 1257.7 | 1264.0 | 1261.5 | 1283.5 |
| Hellinger | 1225.3 | 1243.7 | 1251.1 | 1225.9 |

average the last 10 evaluation among seed 1, 2, and 3 for each environment. Finally, we sum up all the average results from each environment to determine the universal hyper-parameters.

First, we tried to find a suitable $\mathcal{C}_{target}$ value by selecting two different $\tau$ values and three different $\mathcal{C}_{target}$ values and determine which $\mathcal{C}_{target}$ value could have stable and good performance under two different $\tau$ values for both divergences. We have the results shown in Table 5. It demonstrates that when the $\mathcal{C}_{target}$ is 0.1, Hellinger divergence is not good enough, and when $\mathcal{C}_{target}$ is 0.2, it is not very stable for different $\tau$s. Thus, we choose 0.15 as $\mathcal{C}_{target}$, ensuring that the gap between different $\tau$ values is small and both divergences exhibit relatively good performance.

Then, we attempted to find a universal $\tau$ value when $\mathcal{C}_{target}$ to 0.15, using the same evaluation method as above. The results are presented in Table 6. It demonstrates that when the inverse $\tau$ is 10 or 25, the Hellinger divergence does not have a good performance, and when the inverse $\tau$ is 15 or 20, both divergences could have relatively good performance. Thus, we utilized the performance of reversed KL and Pearson divergence to determine which $\tau$ value should be chosen. For reversed KL, when the inverse $\tau$ is 15, the performance summation is 1291.0, and it's 1231.2 when inverse $\tau$ is 20. For Pearson, the performance summation is 1241.1 when inverse $\tau$ is 15, and it's 1236.1 when inverse $\tau$ is 20. Therefore, we can observe that when $\tau$ is $\frac{1}{15}$, all the $f$-divergences exhibit relatively good performance. In summary, we choose $\tau = \frac{1}{15}$ and $\mathcal{C}_{target} = 0.15$ to be the offline datasets' hyper-parameters.

### B.2 ADDITIONAL RESULTS

Here we show additional results for the online and offline experiments.

Fig.1 shows the training curves of different $f$-divergences in the online setting, in which we can see that Pearson divergence is relatively stable and performs better than other choices.

Table 7 and Fig.2 show additional results the offline setting. Fig.1 show the performance of different $f$-divergences using different datasets, in which we can see that reverse KL performs the best among them. Moreover, the common KL divergence is outperformed by every other choices in terms of overall/total performance.

## C RUNNING TIME

### C.1 ONLINE ENVIRONMENTS RUNNING TIME

Table 8 shows the time spent by each $f$-divergence in the online environments, indicating that the time complexity of all $f$-divergences is similar.

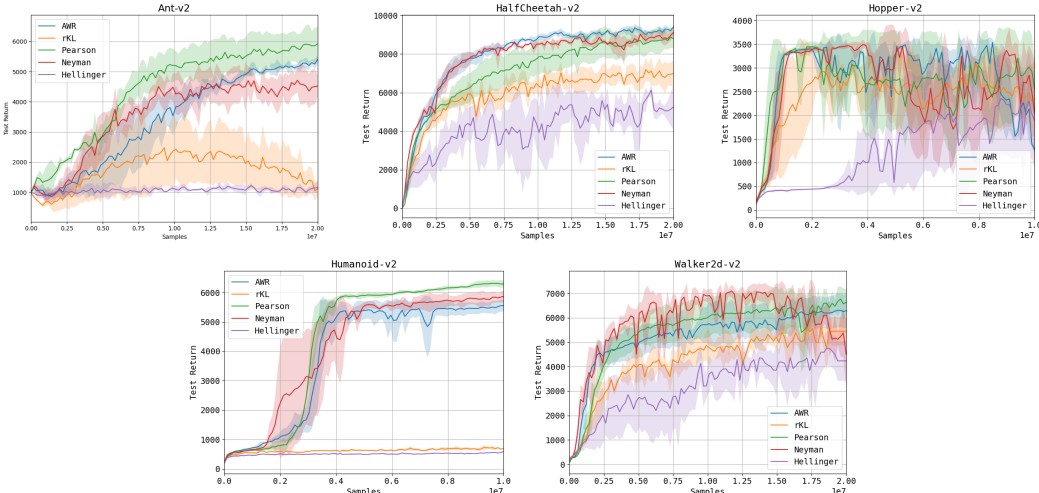

Figure 1: Learning curves of different $f$-divergences in different OpenAI Gym online environments. Results are averaged over seed 1, 2, and 3. Pearson and Neyman divergences have competitive results compare to AWR (which uses KL divergence). However, the reversed KL and Hellinger do not perform very well.

Table 7: $f$-divergences in offline setting, the highest performance is bolded. Ho stands for Hopper, Wa stands for Walker2d, Ha stands for Halfcheetah, r stands for random, m stands for medium, e stands for expert, mr stands for medium-replay, and fr stands for full-replay

| Dataset | IQL (KL) | rKL | Pearson | Neyman | Hellinger |
|---|---|---|---|---|---|
| Ho-r | $9.0 _{\pm 1.3}$ | $8.3 _{\pm 1.0}$ | $7.9 _{\pm 0.1}$ | $\mathbf{9.7} _{\pm \mathbf{0.2}}$ | $8.8 _{\pm 1.8}$ |
| Ho-m | $55.8 _{\pm 4.5}$ | $\mathbf{57.5} _{\pm \mathbf{4.5}}$ | $54.5 _{\pm 5.0}$ | $54.9 _{\pm 5.2}$ | $55.3 _{\pm 3.8}$ |
| Ho-e | $109.5 _{\pm 2.3}$ | $104.9 _{\pm 5.3}$ | $110.3 _{\pm 2.6}$ | $\mathbf{110.7} _{\pm \mathbf{2.8}}$ | $107.5 _{\pm 5.9}$ |
| Ho-m-e | $64.4 _{\pm 34.2}$ | $102.1 _{\pm 8.7}$ | $\mathbf{107.0} _{\pm \mathbf{5.8}}$ | $86.8 _{\pm 15.4}$ | $67.6 _{\pm 18.8}$ |
| Ho-mr | $64.4 _{\pm 7.2}$ | $\mathbf{87.9} _{\pm \mathbf{7.4}}$ | $55.3 _{\pm 6.4}$ | $66.9 _{\pm 6.5}$ | $76.2 _{\pm 9.2}$ |
| Ho-fr | $106.9 _{\pm 0.3}$ | $\mathbf{107.3} _{\pm \mathbf{0.6}}$ | $104.0 _{\pm 0.7}$ | $106.3 _{\pm 1.4}$ | $102.9 _{\pm 5.5}$ |
| Wa-r | $5.5 _{\pm 0.2}$ | $\mathbf{7.2} _{\pm \mathbf{0.7}}$ | $5.8 _{\pm 0.2}$ | $6.5 _{\pm 0.6}$ | $6.5 _{\pm 0.3}$ |
| Wa-m | $\mathbf{78.4} _{\pm \mathbf{4.3}}$ | $72.8 _{\pm 6.7}$ | $77.4 _{\pm 5.7}$ | $75.5 _{\pm 5.7}$ | $75.9 _{\pm 6.5}$ |
| Wa-e | $109.1 _{\pm 0.1}$ | $109.5 _{\pm 1.0}$ | $109.4 _{\pm 0.2}$ | $\mathbf{109.8} _{\pm \mathbf{0.2}}$ | $109.3 _{\pm 1.6}$ |
| Wa-m-e | $\mathbf{109.9} _{\pm \mathbf{0.9}}$ | $109.0 _{\pm 2.2}$ | $108.9 _{\pm 2.0}$ | $109.1 _{\pm 1.8}$ | $108.3 _{\pm 2.9}$ |
| Wa-mr | $\mathbf{72.5} _{\pm \mathbf{10.3}}$ | $72.1 _{\pm 8.5}$ | $62.1 _{\pm 8.1}$ | $69.3 _{\pm 9.4}$ | $68.2 _{\pm 7.8}$ |
| Wa-fr | $86.4 _{\pm 4.6}$ | $92.1 _{\pm 4.4}$ | $86.8 _{\pm 4.9}$ | $94.4 _{\pm 3.3}$ | $\mathbf{94.8} _{\pm \mathbf{3.7}}$ |
| Ha-r | $5.1 _{\pm 1.9}$ | $14.1 _{\pm 5.8}$ | $2.4 _{\pm 0.2}$ | $13.3 _{\pm 6.0}$ | $\mathbf{18.2} _{\pm \mathbf{0.4}}$ |
| Ha-m | $47.4 _{\pm 0.5}$ | $48.3 _{\pm 0.7}$ | $46.3 _{\pm 0.3}$ | $47.6 _{\pm 0.6}$ | $\mathbf{48.5} _{\pm \mathbf{0.4}}$ |
| Ha-e | $95.1 _{\pm 2.6}$ | $94.3 _{\pm 3.4}$ | $\mathbf{95.7} _{\pm \mathbf{0.8}}$ | $95.3 _{\pm 2.5}$ | $90.7 _{\pm 6.8}$ |
| Ha-m-e | $88.5 _{\pm 6.4}$ | $84.0 _{\pm 6.7}$ | $\mathbf{93.4} _{\pm \mathbf{2.6}}$ | $91.3 _{\pm 4.4}$ | $86.5 _{\pm 7.3}$ |
| Ha-mr | $\mathbf{44.0} _{\pm \mathbf{0.8}}$ | $43.8 _{\pm 1.2}$ | $42.3 _{\pm 1.5}$ | $43.1 _{\pm 1.5}$ | $\mathbf{44.0} _{\pm \mathbf{1.2}}$ |
| Ha-fr | $72.4 _{\pm 1.0}$ | $\mathbf{75.8} _{\pm \mathbf{0.7}}$ | $71.6 _{\pm 1.5}$ | $73.5 _{\pm 2.3}$ | $74.5 _{\pm 2.6}$ |
| Total | $1224.3$ | $\mathbf{1291.0}$ | $1241.1$ | $1264.0$ | $1243.7$ |

## C.2 OFFLINE DATASETS RUNNING TIME

Table 8 shows the time spent by each $f$-divergence in the offline datasets, indicating that the time complexity of all $f$-divergences is also similar.

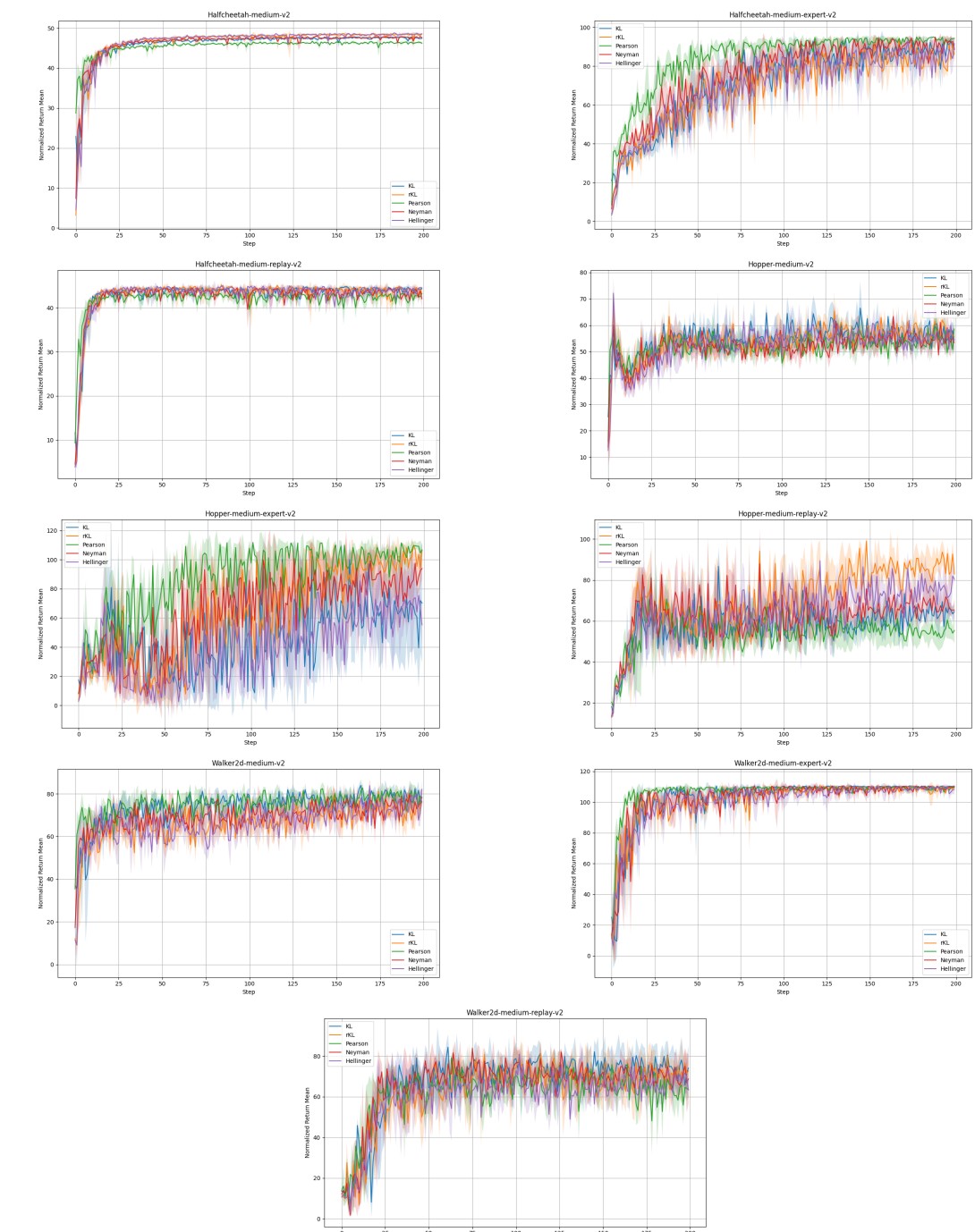

Figure 2: Learning curves of different $f$-divergences in different D4RL offline datasets. Results are averaged over seed 1, 2, and 3. All $f$-divergences have similar performance.

Table 8: Average training time spent by each $f$-divergence in each online environment (hours).

| **Environment** | AWR (KL) | rKL | Pearson | Neyman | Hellinger |
|---|---|---|---|---|---|
| Ant-v2 | 17.02 | 17.29 | 18.08 | 17.45 | 17.52 |
| Halfcheetah-v2 | 14.19 | 14.33 | 13.84 | 14.29 | 13.87 |
| Hopper-v2 | 7.23 | 7.28 | 7.58 | 7.24 | 7.47 |
| Humanoid-v2 | 10.59 | 6.81 | 10.82 | 10.72 | 10.6 |
| Walker2d-v2 | 14.92 | 14.09 | 14.77 | 17.01 | 14.02 |

Table 9: Average training time spent by each $f$-divergence in each offline datasets (hours).

| Dataset | IQL (KL) | rKL | Pearson | Neyman | Hellinger |
|---------|----------|------|---------|--------|-----------|
| Ho-r | 3.7 | 4.13 | 4.27 | 4.57 | 3.96 |
| Ho-m | 3.78 | 3.87 | 3.77 | 3.87 | 3.64 |
| Ho-e | 3.62 | 4.31 | 3.95 | 3.73 | 3.81 |
| Ho-m-e | 3.66 | 3.99 | 3.74 | 3.61 | 3.6 |
| Ho-mr | 3.62 | 3.8 | 3.76 | 4.08 | 3.68 |
| Ho-fr | 3.8 | 3.6 | 3.79 | 3.62 | 3.63 |
| Wa-r | 3.6 | 3.72 | 3.9 | 3.88 | 3.83 |
| Wa-m | 3.86 | 3.55 | 3.7 | 3.67 | 3.62 |
| Wa-e | 3.77 | 3.87 | 3.82 | 3.9 | 3.91 |
| Wa-m-e | 4.08 | 3.88 | 3.78 | 3.83 | 3.92 |
| Wa-mr | 3.84 | 3.67 | 3.56 | 3.77 | 3.53 |
| Wa-fr | 3.72 | 3.71 | 3.77 | 3.61 | 4.08 |
| Ha-r | 3.73 | 4.26 | 4.0 | 3.88 | 3.96 |
| Ha-m | 3.82 | 3.56 | 3.55 | 3.51 | 3.57 |
| Ha-e | 3.81 | 3.97 | 3.73 | 3.69 | 3.81 |
| Ha-m-e | 3.74 | 3.94 | 3.51 | 3.66 | 3.69 |
| Ha-mr | 3.7 | 3.73 | 3.63 | 3.6 | 3.73 |
| Ha-fr | 3.75 | 3.91 | 3.78 | 3.7 | 3.64 |

