# OpenReview forum: "Policy Optimization with $f$-Divergence Regularization"
_ICLR.cc/2026/Conference — Submitted to ICLR 2026_

### Official Review · Reviewer_6ZPL · 2025-10-25

**Soundness:** 2
**Presentation:** 3
**Contribution:** 2
**Rating:** 4
**Confidence:** 4

**Summary:**

This paper proposes a general framework for policy optimization in reinforcement learning based on f-divergence regularization. The authors generalize this idea by introducing f-divergence-regularized Policy Optimization (fRPO), which replaces KL with a broad family of f-divergences (including reverse-KL, Pearson, Neyman, and Hellinger) and provides both theoretical guarantees and potential empirical benefits of doing so.

**Strengths:**

1. **Theoretical soundness**. The derivation of the proposed fRPO framework is theoretically sound. The paper not only provides a rigorous proof of policy improvement but also reveals the underlying connections among many existing methods, offering a unified view of regularized policy iteration.
2. **Evaluation across online and offline settings.** The paper conducts experiments in both online and offline RL settings, demonstrating that the proposed approach can adapt to most mainstream RL paradigms.

**Weaknesses:**

Although the paper presents a clear theoretical motivation and rigorous proofs, the main weaknesses lie in the experimental section.

1. **Limited experiments.**
The experiments are primarily conducted on MuJoCo control tasks, which are relatively simple and may not fully demonstrate the generality of the proposed method. For a framework that claims to be broadly applicable, evaluation on a wider and more diverse set of environments (e.g., Atari) would provide stronger empirical support.
In addition, the number of random seeds appears to be quite limited. Given that reinforcement learning performance can vary significantly across seeds, the reported results—both in online and offline settings—are not entirely convincing without variance measures or statistical significance tests.

2. **Choice and performance of baselines.**
Some baseline methods, such as PPO in the online experiments, exhibit unexpectedly low performance, which raises concerns about the fairness or correctness of the comparison.

3. **Algorithmic empirical performance.**
Moreover, in several tasks the performance improvements from replacing KL with other f-divergences are relatively marginal (even though the paper does not explicitly claim large gains). This observation partially undermines the practical motivation for adopting fRPO: **if the performance difference is small, not guaranteed, and not clearly explained**, it remains unclear why one should prefer fRPO over existing KL-based approaches.

**Questions:**

1. Could the authors provide insights—perhaps in terms of exploration–exploitation balance, policy conservatism, or distributional asymmetry—that explain **how the choice of f affects the behavior and performance of the algorithm across different environments**?

I believe this is likely the core issue of the paper. If the authors could provide a clear explanation—and ideally, supporting experiments—the work would become much more complete and compelling.

---

> ### Author Response · Authors · 2025-11-25
> **Response to Reviewer 6ZPL**
>
> We thank the reviewer for recognizing our theoretical soundness with a rigorous proof and diverse evaluation across online and offline settings. We provide point-to-point responses to the questions as follows:
>
> **Weakness 2**: Choice and performance of baselines. Some baseline methods, such as PPO in the online experiments, exhibit unexpectedly low performance, which raises concerns about the fairness or correctness of the comparison.
>
> **Response**: Several of the results, including PPO, are directly cited from prior papers. Moreover, the PPO results are aligned with the benchmark results from OpenAI Spinning Up website (https://spinningup.openai.com/en/latest/spinningup/bench.html).
>
> **Q1**: Could the authors provide insights—perhaps in terms of exploration–exploitation balance, policy conservatism, or distributional asymmetry—that explain how the choice of f affects the behavior and performance of the algorithm across different environments?
>
> **Response**: Please also see our common response to all reviewers for more details. Based on the result tables in our work, divergences with stronger mode-covering tendencies generally perform better in online settings, while those with stronger mode-seeking tendencies achieve better results in offline settings. Therefore, for online environments, it is preferable to use Pearson or KL divergences in our method, whereas for offline settings, Neyman or reverse KL divergences are typically more suitable.

---

> > ### Comment · Reviewer_6ZPL · 2025-11-26
> >
> > We thank the authors for the discussion regarding the weaknesses.
> >
> > 1. Regarding PPO, the observed performance may stem from biases introduced by the chosen baseline and implementation. In practice, PPO is widely regarded as the go-to RL algorithm in robotics and can easily achieve strong performance on standard MuJoCo locomotion tasks (e.g., Humanoid). I encourage the authors to cross-check with other implementations—such as RLlib (https://docs.ray.io/en/latest/rllib/index.html) or the SB3-based training pipeline (https://github.com/DLR-RM/rl-baselines3-zoo)—which can yield significantly better results.
> >
> > 2. I appreciate the authors’ analysis and clarification, and I find the explanation reasonable.

---

### Official Review · Reviewer_W1rt · 2025-10-31

**Soundness:** 3
**Presentation:** 2
**Contribution:** 2
**Rating:** 6
**Confidence:** 4

**Summary:**

The paper proposes f-divergence–regularized policy optimization (fRPO), a policy-iteration framework that uses general f-divergences as the regularizer in the policy-improvement step. It derives a closed-form target policy and a practical neural-network training recipe, and proves a monotonic-improvement bound for the theoretical update. Experiments in onlineand offline settings show competitive and in some cases superior performance compared to KL-based baselines.

**Strengths:**

The paper provides sufficient theoretical analysis, including a clear derivation and improvement guarantee.

The experimental results are rich, evaluating several f-divergence choices against competitive baselines.

**Weaknesses:**

1. **Motivation feels weak**:
While I understand that different environments may benefit from different regularizers, the paper does not clearly articulate the specific problem fRPO solves. For example, TRPO addresses large on-policy update steps that make fresh rollouts unreliable; PPO offers a simplified, widely usable alternative. In contrast, this paper doesn’t pinpoint a concrete failure mode or practitioner pain point that fRPO uniquely resolves. Given that PPO and SAC are standard in robotics and GRPO is popular in LLM training, it would be more informative to explain and demonstrate how fRPO helps in practice beyond reporting score deltas.

2. **Formulation and hyperparameter selection**:
The empirical results suggest mixed outcomes across different f-divergences, and choosing among them feels similar to tuning PPO/SAC hyperparameters, and fRPO also introduces additional hyperparameters. From users' perspective, it’s unclear why one would invest extra tuning effort here rather than further tuning PPO or SAC. It would be much more useful if the paper provided guidance for selecting the divergence and defaults, such as simple heuristics tied to application and recommended hyperparameter ranges.

**Questions:**

My questions are link to the weaknesses I raise, i.e.

1. What concrete problem does the proposed method solve? In other words, what specific performance capability or apllication does fRPO enable that existing methods do not?

2. How should practitioners choose the f-divergence for a given use case? Is there a principled selection rule or practical guidance with defaults for different regimes?

---

> ### Author Response · Authors · 2025-11-25
> **Response to Reviewer W1rt**
>
> We thank the reviewer for recognizing our theoretical analysis, a clear derivation, and rich experimental results. We provide point-to-point responses to the questions as follows:
>
> **Q1**: What concrete problem does the proposed method solve? In other words, what specific performance capability or application does fRPO enable that existing methods do not?
>
> **Response**: Our results show that fRPO achieves strong and consistent performance in both online and offline settings. First, this indicates that our method is versatile and can be applied to either setting without modifying its overall framework. Second, because different f-divergences exhibit varying degrees of mode-covering (exploration) and mode-seeking (conservativeness/exploitation), and because different settings prefer different strengths, our method can easily adapt to the requirements of each environment or dataset. By simply switching the underlying divergence, fRPO can adjust the balance between exploration and conservativeness to better fit the task at hand. Please also see our common response to all reviewers for more details.
>
> **Q2**: How should practitioners choose the f-divergence for a given use case? Is there a principled selection rule or practical guidance with defaults for different regimes?
>
> **Response**: Based on the result tables in our work, divergences with stronger mode-covering tendencies generally perform better in online settings, while those with stronger mode-seeking tendencies achieve better results in offline settings. Therefore, for online environments, it is preferable to use Pearson or KL divergences in our method, whereas for offline settings, Neyman or reverse KL divergences are typically more suitable.

---

### Official Review · Reviewer_gmb4 · 2025-11-01

**Soundness:** 3
**Presentation:** 3
**Contribution:** 2
**Rating:** 2
**Confidence:** 3

**Summary:**

The paper proposes an $f$-divergence–regularized policy-optimization framework (“fRPO”), where each iteration computes a closed-form, non-parametric target policy by maximizing a standard advantage surrogate penalized by a general $f$-divergence to the current policy, then projects to a neural policy by a forward-KL fit. Theoretically, it gives a TRPO-style monotonic improvement inequality for arbitrary $f$-divergences by relating total variation to $f$-divergences. Empirically, it shows comparable performance to KL-based methods (TRPO/PPO/AWR, etc.) with improvements on certain choices of $f$.

**Strengths:**

1. The paper generalizes KL-regularized policy improvement to arbitrary f-divergences and derives a closed-form target policy.
2. Broadening the choice of f-divergence yields some empirical gains.

**Weaknesses:**

1. While this work extends regularization to arbitrary f-divergences directly in policy space, givent that Belousov & Peters already generalized trust regions to f-divergences at the occupancy (state–action distribution) level and highlighted the benefits of $f$-divergence. Hence, the novelty feels limited.
2. As the main theoretical motivation and contrast, Belousov & Peters is not included as an experimental baseline, leaving the advantage of deriving in policy space (vs. occupancy space) unclear.
3. The lower-bound, TRPO-style improvement guarantee is standard for this family of methods and appears relatively straightforward and incremental.
4. Other related f-divergence based RL works [1-2] are not discussed in the paper.

[1] Agarwal et al. f-Policy Gradients: A General Framework for Goal-Conditioned RL using f-Divergences. NeurIPS 2023

[2] Gong et al. The f-Divergence Reinforcement Learning Framework. 2022

**Questions:**

How does the computational cost and complexity of the closed-form f-divergence–regularized update followed by a forward-KL projection compare with the baselines?

---

> ### Author Response · Authors · 2025-11-25
> **Response to Reviewer gmb4**
>
> We thank the reviewer for recognizing our contributions of generalizing the KL divergence to general f-divergences and the empirical gains. We provide point-to-point responses to the questions as follows:
>
> **Weaknesses 1 & 2**: While this work extends regularization to arbitrary f-divergences directly in policy space, given that Belousov & Peters already generalized trust regions to f-divergences at the occupancy (state–action distribution) level and highlighted the benefits of f-divergence. Hence, the novelty feels limited. As the main theoretical motivation and contrast, Belousov & Peters is not included as an experimental baseline, leaving the advantage of deriving in policy space (vs. occupancy space) unclear.
>
> **Response**: Our method is fundamentally different from Belousov & Peters (2019) in several ways. Theoretically, we offer a monotonic improvement guarantee, showing that the new target policy is guaranteed to be a better policy in terms of maximizing expected return, while their work did not provide such a guarantee. Empirically, the experiments conducted in their paper were limited to simple environments with finite state and action spaces, while our experiments were conducted in much harder environments with continuous state and action spaces. In general, modelling the state-action distribution of a policy, especially over continuous state and action spaces, is very challenging and there is no easy way to do it in their approach. Therefore, their approach is not comparable without a significant and non-trivial modification. This actually signifies a substantial contribution to our approach, making general f-divergence applicable to continuous problems.
>
> **Weakness 3**: The lower-bound, TRPO-style improvement guarantee is standard for this family of methods and appears relatively straightforward and incremental.
>
> **Response**: While the mathematical techniques may be considered standard, their application to the context of general f-divergence brings significant value. Innovation is not always about reinventing the wheel, but about applying well-established tools in novel ways that lead to meaningful results.
>
> **Weakness 4**: Other related f-divergence based RL works [1-2] are not discussed in the paper.
>
> **Response**: We thank the reviewer for providing additional references. For these two papers, we highlight the differences as follows:
> - For Agarwal et al. (2023), it focuses on goal-conditioned reinforcement learning. The f-divergence is used to measure the difference between the state-visitation distribution of goal-achieving trajectories and that of the policy’s trajectories. The objective is to minimize the f-divergence between the distribution induced by the policy and the distribution induced by the goal. This is fundamentally different from our approach as we apply f-divergence to the learning policy and its previous iterate, with theoretical improvement guarantee.
> - For Gong et al. (2022), it formulates an optimization problem that minimizes the divergence between the new policy and a Boltzmann policy derived from the value function of the old policy, which is different from ours in that we do not use the Boltzmann policy as the target. Instead, we learn the target policy by minimizing an f-divergence-regularized objective, and construct a learning objective by minimizing the KL divergence towards that target. This leads to a much easier optimization problem instead of a saddle-point problem in Gong et al. (2022), which is known to be difficult to solve in practice.
>
> We will incorporate discussions of both papers in the revised version.
>
> **Question**: How does the computational cost and complexity of the closed-form f-divergence–regularized update followed by a forward-KL projection compare with the baselines?
>
> **Response**: Please note that we do not have an explicit projection step in our implementation. Instead, the projection step provides an easily implementable objective, suitable for training even for continuous problems via gradient descent. The runtime of our method is similar to that of the AWR/IQL, as shown in the newly added tables in the appendix (Tables 8 and 9). They show the time spent on each environment and dataset across different f-divergences. The results indicate that the computational cost is comparable to that of AWR/IQL.

---

### Official Review · Reviewer_C3bT · 2025-11-01

**Soundness:** 2
**Presentation:** 2
**Contribution:** 1
**Rating:** 2
**Confidence:** 4

**Summary:**

Overall, the paper provides a well-structured and mathematically consistent formulation of policy-regularized reinforcement learning that generalizes TRPO to an arbitrary f--divergence. The authors derive a monotonic policy improvement bound under standard convex optimization assumptions, propose a closed-form target policy using convex conjugates, and develop a practical algorithm (fRPO) that is applicable to both online and offline settings.

Empirically, the method is evaluated on MuJoCo and D4RL benchmarks, demonstrating that non-KL divergences (e.g., Pearson, Neyman) can achieve competitive or superior performance to KL-based baselines such as TRPO, PPO, and AWR.

However, while the theoretical development is sound and the presentation clear, the theoretical novelty appears limited and the empirical gains seem relatively marginal.

**Strengths:**

The paper offers a coherent and mathematically sound formulation of a "general" policy-regularized reinforcement learning framework, though certain aspects could be further clarified or strengthened. The main strengths of the paper are summarized below, and I will ask for specific clarifications in the question section later.

* It derives the monotonic improvement bound following the TRPO [1] framework, and the extension to an arbitrary $f$-divergence is logically valid under the standard convex optimization framework.

* The use of convex conjugate analysis and the resulting dual representation of the policy update are clearly written, showing solid understanding of policy regularization through convex optimization.

* The overall theoretical exposition is clean. The notation and derivations align with standard results in regularized RL (e.g., online: TRPO [1], and offline: SQL [2]).

*Reference*

[1] John Schulman "Trust Region Policy Optimization" Proceedings of the 32nd International Conference on Machine Learning.

[2] Xu, Haoran, et al. "Offline RL with No OOD Actions: In-Sample Learning via Implicit Value Regularization." The Eleventh International Conference on Learning Representations.

**Weaknesses:**

* Lack of Novel Theoretical Contribution (Online RL)

   * The paper extends the monotonic improvement guarantee of TRPO [1] by replacing the KL divergence with a general $f$-divergence. While this extension is mathematically correct, it represents a straightforward generalization rather than a fundamentally new theoretical insight. The proof closely follows the original TRPO derivation, with no substantial change in assumptions, bounding techniques, or underlying theoretical implications.

* Similarity to SQL [2] (Offline RL)

   * The paper applies a behavior-regularized reinforcement learning framework to offline RL, but the resulting convex optimization formulation and loss function appear almost identical to those used in Sparse Q-Learning (SQL) [2].

* Limited Empirical Significance

   * Although the paper reports performance improvements in both online (Table 2) and offline (Table 3) settings, the reported gains over strong baselines such as TRPO, PPO, SAC, and IQL appear relatively marginal. The differences seems to be within the range of variance typically observed in these benchmarks, making it unclear whether the improvements are statistically significant or practically meaningful.
   * In addition, the paper does not provide confidence intervals or any measure of statistical significance for the reported results. This omission makes it difficult to assess whether the observed performance differences are consistent or simply due to random fluctuations across seeds.

**Questions:**

* Q1. Effect of Different *$f$*-Divergences on Policy Characteristics

The paper replaces the KL divergence with a general f-divergence, which is theoretically correct. However, it remains unclear how this substitution affects the qualitative behavior or characteristics of the learned policy. For instance, do certain divergences encourage more conservative, exploratory, or stochastic policies compared to KL? In addition, could the authors elaborate on how these differences translate into practical advantages in (online/offline) RL settings?

* Q2. Could the authors clarify how the proposed formulation fundamentally differs from SQL[2], both in terms of theoretical motivation and algorithmic structure?

* Q3. Statistical Significance of Reported Results

The experimental results in Tables 2 and 3 are presented without any indication of statistical significance, such as confidence intervals or standard deviations across random seeds. Could the authors provide such information? Including these measures would strengthen the empirical claims and help assess whether the proposed method’s improvements are statistically meaningful.

* Q4. Minor Question Regarding Mathematical Expression

In the paragraph right above Section 3.3, could the authors clarify what it means to “approximate” *$D_f(\pi_k|\pi_k)$*? As currently written, the expression of it seems to evaluate to zero in that context, so it is unclear why an approximation is needed. This may be a typographical or notational issue, and a clarification would be appreciated.

---

> ### Author Response · Authors · 2025-11-25
> **Reponse to Reviewer C3bT**
>
> We thank the reviewer for recognizing our contributions of theoretical monotonic improvement guarantee and clear writing of algorithm derivations. We provide point-to-point responses to the questions as follows:
>
> **Q1**: The paper replaces the KL divergence with a general f-divergence, which is theoretically correct. However, it remains unclear how this substitution affects the qualitative behavior or characteristics of the learned policy. For instance, do certain divergences encourage more conservative, exploratory, or stochastic policies compared to KL? In addition, could the authors elaborate on how these differences translate into practical advantages in (online/offline) RL settings?
>
> **Response**: Please refer to our common response to all reviewers.
>
> **Q2**: Could the authors clarify how the proposed formulation fundamentally differs from SQL [2], both in terms of theoretical motivation and algorithmic structure?
>
> **Response**: Compared to Xu et al.(2023), the differences are listed as follows:
> - *Theoretical motivation*: We derived our algorithm from the online setting in which the policy is iteratively optimized from its previous iterate, while SQL was derived from an offline setting where the policy is optimized to not deviate too much from a fixed behaviour dataset. In addition, we ensure monotonic improvement, which is novel compared to SQL.
> - *Algorithm structure*: We use a general clipping scheme applicable to different choices of $f$, while they (may) use an approximation in some cases (specifically for the SQL variant).
> - These distinctions are important and we will include them in the revised version.
>
> **Q3**: The experimental results in Tables 2 and 3 are presented without any indication of statistical significance, such as confidence intervals or standard deviations across random seeds. Could the authors provide such information?
>
> **Response**: We updated Tables 2 and 3 to include standard deviations. We performed a one-sided t-test with the hypothesis that the mean performance of the other divergence is higher than that of the KL divergence. If the resulting p-value is ≤ 0.05, we highlight the entry in red to indicate that the alternative divergence’s mean performance is significantly higher than that of the KL divergence for that environment or dataset. We can see that using other divergences can outperform the KL divergence in roughly half of the instances, showing that they can be advantageous in a significant subset of situations.
>
> **Q4**: In the paragraph right above Section 3.3, could the authors clarify what it means to “approximate” D_f(\pi_k | \pi_k)? As currently written, the expression of it seems to evaluate to zero in that context, so it is unclear why an approximation is needed. This may be a typographical or notational issue, and a clarification would be appreciated.
>
> **Response**: Thank you for pointing out this typo. One of the $\pi_k$ should be $\pi$. We have corrected it in the revised version.

---

> > ### Comment · Reviewer_C3bT · 2025-11-27
> >
> > I appreciate the authors for providing detailed results in Table 2 and Table 3. I agree with the claim that, within fRPO, mode-covering *$f$*-divergences may perform better in online settings, while mode-seeking ones can be advantageous in offline cases.
> >
> > **Online domain**
> >
> > In the online domain, the results show that the proposed method does not fall significantly behind SAC, considering the standard deviations. However, SAC still achieves superior average performance overall, likely due to the exploration effect of the entropy regularization term. While the mode-covering property in fRPO introduces an interesting exploration effect, it appears less effective than explicit entropy-based exploration.
> >
> > **Offline domain**
> >
> > Regarding the Offline RL setting, the authors mention two main differences from SQL. Although the theoretical motivation based on monotonic improvement from the previous iterate is valid, Algorithm 1 suggests that, in the offline case, the data-collecting policy remains fixed, making the reasoning essentially similar to SQL. Furthermore, the clipping operation, equivalent to applying *$\max(0,\cdot)$* to $(f')^{-1}$, has already been explored in prior literature.
> >
> > In conclusion, while the exploration effect of the mode-covering *$f$*-divergence is interesting, the mode-seeking–based offline domain seems to offer limited novelty, and therefore I would like to maintain my current score.

---

### Author Response · Authors · 2025-11-25
**Common Response**

We thank all reviewers for their insightful comments and valuable feedback. Here, we provide an additional insight for using general f-divergence from the perspective of mode-seeking (exploitation) versus mode-covering (exploration). All revisions are highlighted in red in the paper (statistical testing and discussions on the choice of f-divergence in the main text, and runtime tables in the appendix).

As noted by Li and Farnia (2023), different f-divergences exhibit different tendencies towards mode-seeking (concentrating on a high value region or mode) or mode-covering (attempting to cover multiple modes). Mode-covering encourages greater exploration, whereas mode-seeking promotes more conservative behaviour by exploiting regions that appear promising. As illustrated in Figure 1 of Li and Farnia (2023), Pearson divergence shows the strongest mode-covering tendency, followed by KL divergence. In contrast, reverse KL and Neyman divergences exhibit increasing degrees of mode-seeking behaviour, with Neyman divergence being the most mode-seeking overall. Hellinger divergence lies in the middle, exhibiting a balanced degree of mode-covering and mode-seeking.

In our context, our empirical results show that mode-covering divergences tend to perform well in online environments (Pearson and KL), while mode-seeking divergences generally achieve good performance in offline environments (reverse KL and Neyman). These findings suggest that, in OpenAI Gym online environments, divergences that encourage exploration tend to yield better performance. In contrast, for D4RL offline datasets, divergences that promote conservativeness perform well. This highlights a key advantage of using f-divergences and our model: the choice of divergence can be adapted to the characteristics of the environment or dataset—for example, online vs. offline, or exploration vs. conservativeness. Our method allows the f-divergence to be switched easily, enabling practitioners to adjust the balance between mode-covering and mode-seeking to best fit the task at hand.

Reference:
- Li, C.T. and Farnia, F., 2023, April. Mode-seeking divergences: theory and applications to GANs. In International Conference on Artificial Intelligence and Statistics (pp. 8321-8350). PMLR.

---

### Meta-Review · Area_Chair_DUwJ · 2026-01-07

**Summary:**

This paper presents fRPO, a framework that generalizes policy optimization by replacing the standard KL-divergence with a broader family of f-divergences. The authors provide a theoretical monotonic improvement guarantee and derive a closed-form target policy. While the reviewers appreciated the mathematical soundness and the extension to both online and offline RL, the final consensus leaned towards rejection. The primary concerns included: (1) Limited Theoretical Novelty: Reviewers noted that the derivation closely follows the TRPO framework and the offline formulation is very similar to existing methods like SQL. (2) Marginal Empirical Gains: The performance improvements over standard baselines like PPO and SAC were viewed as incremental or within the range of standard variance. (3) Lack of Practical Motivation: It remains unclear which specific "pain points" this method solves that highly tuned versions of PPO or SAC do not already address.

**Reviewer Concerns:**

The authors updated the results with standard deviations and t-tests to show where non-KL divergences outperformed the baseline, which was aimed at addressing the concern about the marginal gain of the empirical performance. The authors also provided a common response explaining the "mode-seeking" vs. "mode-covering" behaviors of different divergences in online vs. offline settings, which partially addressed the motivation of the proposed framework. However, the empirical gains were still marginal in many cases. Reviewer 6ZPL also noted that the PPO baseline performance seemed lower than what is typically achieved by state-of-the-art implementations. Another major unresolving concern is about the novelty of the proposed framework, which largely follows TRPO. The explanation on "mode-seeking" vs. "mode-covering" behaviors of different divergences is interesting. But this might require significantly more well-designed experiments to support such claims.

**Reviewer Scores:**

Reviewer C3bT: 2 (Reject). Likely would stay at 2; the rebuttal didn't overcome the perceived lack of novelty.

Reviewer gmb4: 2 (Reject). Likely would stay at 2; felt the work was incremental.

Reviewer W1rt: 6 (Marginally Accept). Might stay at 6 or dip to 4, as they acknowledged the theory is sound but felt the motivation was "weak."

Reviewer 6ZPL: 4 (Marginally Below). Might stay at 4; appreciated the mode-seeking explanation but remained skeptical of the empirical setup.

---

### Decision · Program_Chairs · 2026-01-26

Reject